# Validation of *Bos taurus* SNPs for Milk Productivity of Sahiwal Breed (*Bos indicus*), Pakistan

**DOI:** 10.3390/ani14091306

**Published:** 2024-04-26

**Authors:** Asma Younis, Imtiaz Hussain, Syeda Nadia Ahmad, Amin Shah, Iram Inayat, Muhammad Ali Kanwal, Sadia Suleman, Muhammad Atif Kamran, Saima Matloob, Khawaja Raees Ahmad

**Affiliations:** 1Department of Zoology, University of Sargodha, Sargodha 40100, Pakistan; asma.younis92@gmail.com (A.Y.); iram.inayat@uos.edu.pk (I.I.); atifpetangi45@gmail.com (M.A.K.); saimamatloob42@gmail.com (S.M.); 2Department of Animal Sciences, University of Sargodha, Sargodha 40100, Pakistan; 3Department of Zoology, University of Chakwal, Chakwal 48800, Pakistan; nadia.ahmad@uoc.edu.pk; 4Department of Botany, University of Sargodha, Sargodha 40100, Pakistan; aminullah.amin@uos.edu.pk; 5Higher Education Department, Government of Punjab, Lahore 40100, Pakistan; sahargul24@gmail.com

**Keywords:** casein, Sahiwal cattle, single nucleotide polymorphism, milk production, Pakistan, PCR

## Abstract

**Simple Summary:**

This present research work was designed for the validation of some SNPs of casein in *Bos indicus* (Sahiwal breed), previously reported for their association with milk yield in *Bos taurus*. This study indicated that these SNPs of casein genes (rs43703010, rs109500451, rs110079521, rs110323127, rs43703015, rs43703016 and rs43703017) significantly affected milk yield in *Bos indicus* of the Sahiwal breed. Animals with the genotype GG, having the SNP α S1 and aA S2 casein genes (rs43703010 and rs109500451), show high milk yield. For the SNPs of the κ casein genes (CSN3) (rs110079521, rs43703016 and rs43703017), animals with genotype AA were high milk producers. For the other two SNPs of the CSN3 gene (rs110323127 and rs43703015), animals with genotypes GG and CC produced a high milk yield. In light of these results, animal breeders may be able to make a more precise selection of the animals of *Bos indicus* (Sahiwal breed) with high milk yield.

**Abstract:**

The aim of the present study was the validation of the already reported *Bos taurus* SNPs in the Sahiwal breed. A total of nine SNPs of the casein gene were studied. Out of nine, seven *Bos taurus* SNPs of casein protein genes were found to be significantly associated with milk productivity traits. The genomic DNA was extracted from the mammary alveolar endothelial cells of a flock of 80 purebred Sahiwal lactating dams available at Khizrabad Farm near Sargodha. New allele-specific primers were designed from the NCBI annotated sequence database of *Bos taurus* to obtain 100 nt-long PCR products. Each dam was tested separately for all the SNPs investigated. Animals with genotype GG for the SNPs rs43703010, rs10500451, and 110323127, respectively, exhibited high milk yield. Similarly, animals with genotype AA for the SNPs rs11079521, rs43703016, and rs43703017 showed high milk yield consistently. For the SNP rs43703015, animals with genotype CC showed high milk productivity. These above-mentioned SNPs have previously been reported to significantly up-regulate casein protein contents in *Bos taurus*. Our results indicated SNPs that significantly affect the milk protein contents may also significantly increase per capita milk yield. These finding suggest that the above-mentioned reported SNPs can also be used as genetic markers of milk productivity in Sahiwal cattle.

## 1. Introduction

Pakistan is an agricultural country and among the major milk producers in the world [1,2]. The Sahiwal is one of the most important dairy breeds in Pakistan, known for its high milk yield and ability to adapt to hot and humid environments [3,4]. Milk yield is a critical economic trait in the dairy industry [5]. To enhance milk yield, animal breeders had been selecting animals based on their phenotypes for centuries. However, using conventional breeding in livestock production is time-consuming and does not cover all aspects of genetic variation [6].

Molecular genetic techniques are the emerging trend of the age [7]. Genes affecting animal performance traits can efficiently improve the animals’ breeding value [8]. Current development in molecular knowledge and techniques has discovered many genetic polymorphisms at DNA level, and they are correlated with animal performance. Single-nucleotide polymorphisms (SNPs) have gained popularity as markers of phenotypic variations among the individuals of a population [9]. Over the past few years, SNPs have been in the spotlight of molecular genetics research due to their abundance in a given genome, ease of replication in laboratory conditions, and association with specific traits [10]. For the selection of genetically superior cows with high milk yield, it is very important to identify the genotypic polymorphism of the genes affecting the milk quality and quantity [11].

In this study, several specific SNPs known to be associated with milk productivity in Bos taurus cattle were focused on. These SNPs include rs43703010 (A/G), rs109500451 (A/G), rs110079521 (C/A), rs110323127 (A/G), rs43703015 (C/T), rs43703016 (A/C), and rs43703017 (A/G). They were studied in various breeds, including Holstein, Jersey, and Angus. These SNPs have been found to affect milk traits such as milk yield, fat content, protein content, lactose concentration, and somatic cell count.

Bovine milk is a chief source of protein in human diets. Estimating the amount of protein content in milk is very effective way to find out the nutritive value of the milk [12]. Casein and whey are two gene families that mainly control the protein properties of dairy cows [11]. Various studies have reported that milk protein variants also affect milk productivity [13]. 

Casein genotype has diverse effects on milk protein [14]. Four casein genes (αS1-, αS2-, β- and κ-CN), which account for more than 75% of total bovine milk proteins, are encoded in a tightly linked 250-kb cluster on BTA 6 [15]. The genotype at various loci of *CSN1S1*, *CSN1S2*, and *CSN3* are found to affect milk composition and milk yield [14]. The casein genes in bovines are polymorphic [16]. They are responsible for affecting the qualities of the milk that provide the suckling infant with calcium, amino acids, and phosphate. They also affect the cheese-making properties and milk-production traits [17].

Incoming information indicates that identification of SNPs as genetic markers of milk productivity can pay off hugely in terms of capital and national health. Moreover, the study of breeds, using molecular techniques, is very important and useful for their characterizing [18,19]. Preserving genetic diversity in animal species necessitates the effective implementation of conservation strategies and sustainable management plans. These plans should be informed by comprehensive knowledge of population structures, encompassing genetic diversity resources within and between breeds [20,21]. Genetic diversity is an essential element for genetic improvement, preserving populations, evolution, and adapting to variable environmental situations [22]. On the other hand, determination of gene polymorphism is important in farm animal breeding [23,24] in order to define the genotypes of animals and their associations with productive, reproductive, and economic traits [25].

However, most of this work has remained confined to studies on the Holstein Friesian breed of exotic cattle (*Bos taurus*), while unfortunately little is known about the indigenous milk-producing Sahiwal breed. 

Considering this information, the present research work aim was to select the *Bos taurus* SNPs that have been reported to be significantly associated with milk productivity and to study the Sahiwal cattle genome for the presence these SNPs, and to further validate their association with milk productivity in this indigenous cattle breed.

## 2. Materials and Methods

### 2.1. Sample Collection and DNA Extraction

Sahiwal breed milk samples were collected from 80 lactating cows at Khizarabad Farm, Sargodha, Punjab, Pakistan. Samples were collected from individual animals using sterile plastic bottles. The samples were collected aseptically by trained personnel following standard procedures to avoid contamination. Before milking, the udder was cleaned with a sterile cloth to avoid contamination with bacteria from the teat canal. Approximately 20 mL of milk was then collected from each animal and transferred immediately to sterile plastic bottles. The bottles were then labeled with a unique identification number and transported on ice to the laboratory for further analysis. 

The milk yield data of each animal was obtained from the farm records. The average daily milk yield for each animal was calculated by dividing the total milk yield for a given period by the number of days in that period. The milk yield data were recorded by trained personnel at the farm using a standardized milk measurement system. The data collected included the animal identification number, date of milk measurement, and the amount of milk produced.

The DNA extraction methods suggested by Pakrosa et al. (2016) [26] was employed to extract DNA from mammary alveolar endothelial cells present in milk samples.

### 2.2. DNA Measurement and Dilution

The concentration and purity of extracted DNA were determined using a picoDrop spectrophotometer (PicoDrop 200, Cambridgeshire, UK), which was provided courtesy of Agriculture College, University of Sargodha. The DNA was diluted to a working concentration of 50 ng/μL based on the DNA estimation obtained from the PicoDrop measurements, and diluted DNA was stored at −20 °C until for further analysis.

### 2.3. Primer Designing

To detect the CSN SNPs in the genome of *Bos indicus* (Sahiwal breed), an allele-specific PCR method was employed [27]. The primers were designed (picked) from the Bos taurus annotated genome present on the NCBI database. For alternative alleles of a given SNP (for instance, A/G), two alternative primers were designed (selected), either on the forward or the reverse strand, in such a way that one of the primers may end up with A while the other ends up with G (Table 1). The other primer (of the polarity inverse to the above-mentioned SNP-specific alternative in the two primers) was selected in such a way that, when it is used with either of the SNP-specific primers, it may produce 100 nt-long PCR products. It is easily understandable that the SNP-specific primers would work for PCR only in the presence of their specific nucleotide on the genomic DNA (e.g., T for the above-mentioned A primer and C for the above-mentioned G primer). For the above-mentioned (A/G) SNP, a given individual may have one of three possible genotypes (AA, GG, and AG). To ascertain the genotype of all individuals under study, two separate PCR reactions were employed, one with the SNP-specific primer A and the other with the SNP-specific primer G. For a given animal, if both PCR products were from the same individual, the animal must be declared heterozygous for the given SNP. Contrarily, if only one PCR product forms (for example, the PCR revealed only primer A), the individual animal must be treated as homozygous for the A allele or vice versa. Which of the two PCRs is actually positive is easily identifiable with gel electrophoresis (Figure 1).

### 2.4. PCR Protocol

PCR was performed using a 2X Taq PCR master by Bishop, Burlington, ON, Canada (Cat. No TAQ006D). The PCR contents were mixed as follows: 0.5 μM each forward and reverse primer, 2X PCR master mix 25 μL, template DNA 1 μL (approximately 50 ng), nuclease-free water 23 μL; the total volume was 50 μL. The thermal profile included initial denaturation at 94 °C for 3 min, primer annealing at 55 °C for 30 s, extension for 1 min at 72 °C, and final extension for 10 min at 72 °C [28]. The total number of cycles was 35. The thermal profile used for PCR amplification was nearly identical for all the primers designed. 

Agarose gel (1%) was used to visualize and confirm the results obtained from the AS-PCR procedure. 

### 2.5. Statistical Analysis

#### 2.5.1. Linkage Disequilibrium

Linkage disequilibrium (LD) was measured for the SNPs genotyped within casein genes using the online software SHESIS [29]. The dataset included the genotyping data of *Bos indicus* (Sahiwal breed) for the seven CSN SNPs located on chromosome 6. To perform LD analysis, pairwise LD was calculated between all SNPs in the dataset. The standardized disequilibrium coefficient (D′) and squared correlation coefficient (r^2^) were calculated as a measurement of LD between pairs of SNPs. The LD blocks were defined using a solid spine algorithm.

#### 2.5.2. Haplotype Analysis and Association

To calculate the Hardy–Weinberg equilibrium, the genotype frequencies of the SNPs in the dataset were compared to the expected frequencies under the Hardy–Weinberg equilibrium model. The chi-square statistic and associated *p*-value were calculated to test for deviation from the Hardy–Weinberg equilibrium.

Haplotype construction and frequency determination were performed using genotype data to infer the underlying haplotype phase of the single-nucleotide polymorphisms (SNPs) present in the dataset. The frequency of each inferred haplotype was calculated.

Association analysis with a phenotype (milk yield) was performed using statistical models, such as linear regression. The genotypes and haplotype frequencies were used to test for associations between the haplotypes and the phenotype, and *p*-values were calculated to indicate the strength of the association. These analyses were performed using the software program SNPSTAT [30].

## 3. Results

Initially, nine SNPs (rs110672723, rs43703011, rs43703010, rs109500451, rs110079521, rs110323127, rs43703015, rs43703016 and rs43703017) were studied. Except for rs110672723 and rs43703011 (as their *p* values were non-significant), all SNPs showed suggestive associations with milk productivity. 

The SNP rs43703010 is a missense variant of *CSN1* gene. SNP rs109500451 is an intronic SNP of the *CSN1S2* gene. Five other SNPS (rs110079521, rs110323127 (intronic), rs43703015, rs43703016 and rs43703017 (missense variants)) are located in *CSN3* gene (Table 2). 

The SNPs of *CSN1*, *CSN1S2*, and *CSN3* are in Hardy–Weinberg equilibrium (*p* > 0.05) except the two SNPs rs43703010 and rs110079521. The observed agreement with the HWE suggests limited evidence of significant natural or artificial selection for these SNPs in our studied population of Bos indicus (Table 3). 

### 3.1. Single Locus-Based Analysis

To show the effect of SNP genotype on milk productivity, milk production was investigated for each of the three possible genotypes.

Regarding SNP rs43703010 on the *CSN1* gene, animals with the GG genotype had the highest milk yield (10.09 ± 0.1), followed by those with the AG genotype (9.85 ± 0.13), and the AA genotype had the lowest milk yield (8.03 ± 0.18). This suggests that the GG genotype is associated with higher milk production in animals.

In the case of SNP rs109500451 on the *CSN1S2* gene, animals with the GG genotype had the highest milk yield (10.14 ± 0.17), followed by those with the AG genotype (9.21 ± 0.14), and the AA genotype had the lowest milk yield (7.55 ± 0.27). These results indicate that the GG genotype is more favorable for milk production in animals.

Regarding SNP rs110079521 on the *CSN3* gene, animals with the AA genotype had the highest milk yield (10.33 ± 0.05), followed by those with the AC genotype (8.77 ± 0.11), and the CC genotype had the lowest milk yield (7.6 ± 0.27). This suggests that the AA genotype may be associated with higher milk production in animals, while the CC genotype seems unfavorable for milk production.

In the case of SNP rs110323127 on the *CSN3* gene, animals with the GG genotype had the highest milk yield (9.97 ± 0.08), followed by those with the GA genotype (9.14 ± 0.43), and the AA genotype had the lowest milk yield (7.61 ± 0.17). These results suggest that the GG genotype is more advantageous for milk yield in animals.

Regarding SNP rs43703015 on the *CSN3* gene, animals with the CC genotype had the highest milk yield (10.72 ± 0.06), whereas TT genotype had lower milk yield (9.16 ± 0.12), and the TC genotype had the lowest milk yield (8.43 ± 0.45). These results indicate a potential association between the CC genotype and higher milk production in animals.

The milk yield comparisons for animals with different genotypes of SNP rs43703016 in the *CSN3* gene showed that the AA genotype had the highest milk yield (10.29 ± 0.06), followed by those with the AC genotype (9.07 ± 0.14), and the CC genotype had the lowest milk yield (7.6 ± 0.27). These finding suggest that the AA genotype could potentially be advantageous for milk production in animals.

Based on the analysis of SNP rs43703017 in the *CSN3* gene, it was observed that animals with the AA genotype had the highest milk yield (10.39 ± 0.06), compared to those with the GA genotype (9.86 ± 0.09), and the GG genotype (7.81 ± 0.15), which had the lowest milk yield. These finding suggest that the AA genotype may have a positive association with milk production in animals (Table 4).

### 3.2. Pairwise Linkage Disequilibrium

Figure 2 illustrates the linkage disequilibrium of these CSN genes. 

### 3.3. Haplotype Frequencies and Association with Milk Yield

Table 5 presents the haplotype frequencies of seven SNPs (SNP1-SNP7) on the *CSN1*, *CSN1S2,* and *CSN3* genes among the animal population studied. The most common haplotype was GGAGTAA (frequency of 0.2006), followed by AACATCG (0.1705) and AACGCCG (0.0625). The rarest haplotypes were GGAGCCG (0.0061), GAAGCAG (0.0029), AACACCG (0), and AGCGCCG (0).

The haplotype GGAGTAA was observed twice in the dataset, indicating its high frequency and potential importance for milk production in the population studied. The two occurrences of GGAGTAA had different frequencies, indicating some level of genetic diversity among animals with this haplotype.

In contrast, the haplotypes GACGTCA and AGCGCCG were observed only once in the dataset, indicating their extremely low frequencies in the population studied. These rare haplotypes may have limited impact on milk production in the population, but their presence in the dataset may provide insights into genetic diversity and population structure. Overall, the haplotype frequencies presented in this (Table 4) provide valuable information for understanding genetic variation in the *CSN1*, *CSN1S2,* and *CSN3* genes and its potential association with milk production traits in the population studied.

The results presented in Table 6 provide important insights into the association between specific haplotypes and milk yield in the studied population of *Bos indicus* (Sahiwal breed). The most frequent haplotype observed in the studied population is “GGAGTAA” with a frequency of 0.1964, and it is highly statistically significant (*p* < 0.0001). This haplotype is associated with a higher milk yield. The second most frequent haplotype is “AACATCG” with a frequency of 0.1638, and it is also highly statistically significant (*p* < 0.0001). This haplotype is also associated with a higher milk yield.

The remaining haplotypes exhibited lower frequencies, ranging from 0.0188 to 0.0383 and have *p*-values ranging from 0.01 to <0.0001, indicating varying degrees of statistical significance. Some of these haplotypes, such as “GGAGCAA” and “GGAACAA”, are associated with a lower milk yield.

These results have important implications for the genetic improvement of milk yield in the studied population of the Sahiwal breed.

## 4. Discussion

There are 30 chromosomes in the bovine haploid genome. As milk productivity traits are multifactorial, it appears that almost all the autosomal genes may affect the milk quantity and quality [31]. Among these autosomes, chromosome 6 is of prime importance, as it harbors many QTLs effecting milk productivity.

The results of our study indicate that the SNP rs43703010 in the *CSN1* gene is significantly associated with milk yield in animals. Animals with the GG genotype had the highest milk yield, followed by those with the AG genotype, while the AA genotype was associated with the lowest milk yield. This finding is consistent with previous studies that have reported an association between the *CSN1* gene and milk production in various dairy breeds [32,33].

The *CSN1* gene encodes for casein, which is the major protein component of milk. The protein is synthesized in the mammary gland and secreted in the milk [34]. The rs43703010 gene is a well-studied genetic variant located on the CSN1 gene in cattle and which plays a crucial role in determining the quality and quantity of milk produced by cows [35].

The GG genotype (SNP rs43703010) in our study was associated with the highest milk yield, and this may be due to the fact that the G allele is associated with higher casein expression and milk protein content. This is consistent with other studies that have reported an association between this gene and higher milk production in different dairy breeds [36]. 

The present study investigated the association between SNP rs109500451 in the *CSN1S2* gene and milk yield in a population of dairy cattle. The results revealed that the GG genotype of this SNP was significantly associated with higher milk yield in animals. 

The *CSN1S2* gene encodes for the protein β-casein, which is a major milk protein and plays a crucial role in determining the quality and quantity of milk production [37]. β-casein is responsible for the formation of micelles that transport calcium, phosphorus, and other nutrients to the mammary gland during lactation [38]. Therefore, any variation in the *CSN1S2* gene may affect the production and quality of milk.

The GG genotype of SNP rs109500451 has been associated with higher β-casein concentrations in milk [39]. β-casein concentration has been shown to have a positive correlation with milk yield in dairy cattle [39]. Therefore, it can be inferred that the GG genotype of SNP rs109500451 may enhance milk yield by increasing β-casein concentration in milk.

The present study found that the SNP rs110079521, located in the *CSN3* κ-casein gene, and its genotype AA are positively associated with milk yield. Existing literature has shown that this SNP is significantly associated with milk protein content, particularly k-casein content. A previous study reported that the genotype AA for this SNP was significantly associated with milk protein content and milk k-casein content [40]. The heterozygous condition AC was found to be slightly less effective in enhancing the milk protein content and milk k-casein content.

The association of this SNP with milk yield and milk protein content is biologically plausible, as κ-casein is a major component of milk proteins and plays a significant role in casein micelle structure, stability, and milk coagulation [41]. 

In our study, the GG genotype for SNP rs110323127 on the *CSN3* gene was found to be significantly associated with higher milk yield in Sahiwal cattle. This is in agreement with previous studies that reported a positive correlation between CSN3 and total milk protein concentration [42]. The GG genotype is likely to enhance milk yield by increasing the production of casein proteins, which are the main proteins in milk and play a crucial role in the regulation of milk synthesis [43].

Likewise, the present study identified the SNP rs43703015, located on the *CSN3* gene, as significantly associated with per capita milk yield in studied population of Sahiwal cattle. Specifically, animals with the CC genotype showed a significant increase in milk yield compared to those with the CT or TT genotypes. This is in line with previous studies which have reported this gene’s association with milk quality [44,45] and fat yield [12].

The CC genotype may enhance milk yield by increasing milk fat content [46]. This is likely due to the effect of the SNP on the expression of the *CSN3* gene, which encodes for the protein κ-casein. κ-casein is involved in the formation of micelles that transport fat and other milk components in mammary epithelial cells. Therefore, variations in κ-casein can affect the synthesis and secretion of milk components [47].

Interestingly, previous studies have also reported that animals with the T allele for this SNP negatively affect milk’s lactose percentage. Lactose is an important component of milk and plays a significant role in milk yield. Therefore, the presence of the T allele may negatively impact lactose synthesis and result in decreased milk yield [48].

The present study found a significant association between the SNP rs43703016 located on the *CSN3* gene and milk yield in Sahiwal cattle. The results showed that the CC genotype was significantly associated with a lower milk yield compared to the AC and AA genotypes. Existing literature shows this SNP was significantly associated with milk’s casein index, κ-CN, β-LG index, and protein percentage [49].

Previous studies have reported the association of this SNP with milk yield in various cattle breeds [50]. One of these studies reported that the A allele of the SNP decreased the milk’s κ CN concentration [27]. Another study conducted on Holstein cattle reported that the CC genotype was associated with a lower milk yield compared to the AA genotype, whereas the AC genotype was associated with intermediate milk yield [51]. These findings suggest that this SNP was affecting the milk yield, and its effect may vary across different cattle breeds.

Another SNP, rs43703017, is present on the k-CN gene. In our studied population, Sahiwal dams with the AA genotype for this SNP show significantly high milk yields. Previous studies have indicated that this SNP is highly associated with milk protein percentage in various dairy breeds [52], although some studies on Holstein Friesian populations have reported its negligible effect on milk composition [53]. This parallelism indicates that whenever the quantity of this protein increases, in turn, milk quantity will also increase.

In conclusion, the present study provides evidence for the significant association of the above-mentioned SNPs with milk yield in the studied population of *Bos indicus* (Sahiwal breed). These findings may have practical implications for breeding programs aimed at improving milk yield in this breed. Further studies are needed to elucidate the underlying mechanism by which this SNP affects milk yield.

The study found that certain haplotypes were associated with milk yield in the studied population of Bos indicus (Sahiwal breed). Specifically, the haplotype GGAGTAA was the most common and also had the highest frequency among the haplotypes associated with higher milk yield. This finding is consistent with previous studies that have shown associations between certain haplotypes and milk yield in different cattle breeds [54].

The genetic variations in the studied haplotypes could be affecting the expression and function of genes related to milk production, such as those encoding for milk proteins or enzymes involved in milk synthesis [55]. Therefore, the identified haplotypes could be useful in breeding programs to improve milk yield in the Sahiwal breed of Bos indicus. 

## 5. Conclusions

This is the first study of its nature reporting on casein SNPs’ association with milk productivity in the Sahiwal breed of Pakistan. This study not only identifies the SNPs associated with milk yield but also makes great strides in sire selection. The dairy industry of Pakistan may benefit, as milk quantity can be improved in a shorter time by identifying SNPs. 

## Figures and Tables

**Figure 1 animals-14-01306-f001:**
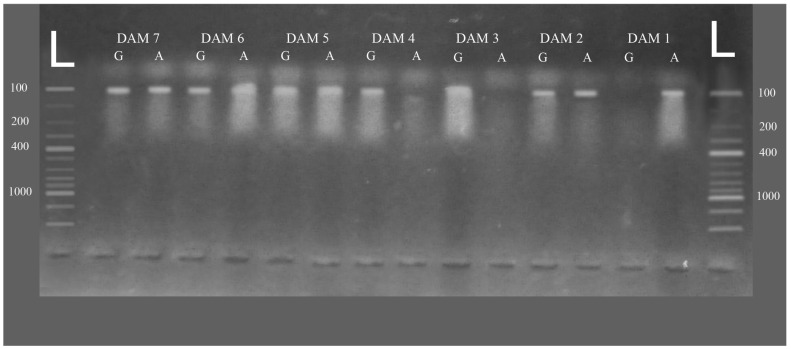
L = 100 bp ladder, Dam 1 = homozygous AA, Dam 2 = heterozygous AG, Dam 3 = homozygous GG, Dam 4 = homozygous GG, Dam 5, 6 and 7 = heterozygous AG respectively.

**Figure 2 animals-14-01306-f002:**
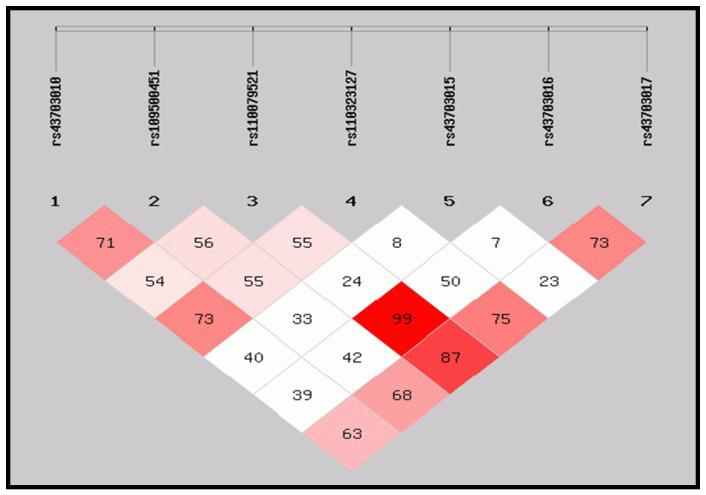
LD of CSN haplotypes and linkage disequilibrium. Pairwise linkage disequilibrium (LD) values (D′) and the haplotype blocks for the seven SNPs on the casein gene. The values within boxes are the pairwise SNP correlation (D′); bright red boxes without numbers indicate complete LD (D′ = 1). The brighter shade of red indicates higher LD.

**Table 1 animals-14-01306-t001:** List of primers.

Sr#	SNP Ref No	Gene		5 > 3
1	rs110672723	*CSN2*	Fwd	GCCTTTTGGCAAGAAA
			Rev	CCACATGACTCATTTCACATCT
			Rev	CCACATGACTCATTTCACATCG
2	Rs43703011	*CSN2*	Fwd	GTTTTGTGGGAGGCTGTTAG
			Fwd	GTTTTGTGGGAGGCTGTTAT
			Rev	TCTTTCCAGGATGAACTCCAG
3	rs43703010	*CSN1S1*	Fwd	ATTACGTTCCACTAGGCAC
			Rev	CAGTGGCATAGTAGTCTTTT
			Re	CAGTGGCATAGTAGTCTTTC
4	rs109500451	*CSN1S2*	Fwd	GGTATAAAACTGATAG
			Rev	GGCTGATACGTTAAGAAATC
			Rev	GGCTGATACGTTAAGAAATT
5	rs110079521	*CSN3*	Fwd	AATCAACCAGATGGATGATA
			Fwd	AATCAACCAGATGGATGATC
			Rev	GTTTTGCCATACATCAACAT
6	rs110323127	*CSN3*	Fwd	CAAGAAGTGGAAGGAAG
			Fwd	CAAGAAGTGGAAGGAAA
			Rev	CAGTATTTGATGTTAAG
7	rs43703015	*CSN3*	Fwd	CCTACAAGTACACCTACCAT
			Fwd	CCTACAAGTACACCTACCAC
			Rev	CTGTGTTGATCTCAGGTG
8	rs43703016	*CSN3*	Fwd	TGTAGCTACTCTAGAAGC
			Fwd	TGTAGCTACTCTAGAAGA
			Rev	CCTTAGAGTATTTAGACCGC
9	rs43703017	*CSN3*	Fwd	GCTTCTCCAGAAGTTATTGAGA
			Fwd	GCTTCTCCAGAAGTTATTGAGG
			Rev	ACCTGCGTTGTCTTCTTTG

**Table 2 animals-14-01306-t002:** Overview of casein gene SNPs selected for present study.

SNP No	Allele	Chromosome	Gene
rs43703010	A/G	6	*CSN1*
rs109500451	A/G	6	*CSN1S2*
rs110672723	A/C	6	*CSN2*
Rs43703011	G/T	6	*CSN2*
rs110079521	C/A	6	*CSN3*
rs110323127	A/G	6	*CSN3*
rs43703015	C/T	6	*CSN3*
rs43703016	A/C	6	*CSN3*
rs43703017	A/G	6	*CSN3*

**Table 3 animals-14-01306-t003:** Genotype and allelic frequency and Hardy–Weinberg equilibrium test of CSN SNPs in *Bos indicus* (Sahiwal breed) (80 animals).

Gene	Position	Locus	Genotype	N	Frequency	Allele	Frequency	Hardy-Weienberg Equilibrium X2 Test
*CSN1*	Fwd	rs43703010	AA AG GG	37 13 30	0.38 0.16 0.46	A G	0.54 0.46	*p* < 0.05
*CSN1S2*	Fwd	Rs109500451	AA AG GG	18 39 23	0.22 0.49 0.29	G A	0.53 0.47	*p* > 0.05
*CSn3*	Fwd	Rs110079521	AA AC CC	32 28 20	0.4 0.35 0.25	A C	0.57 0.42	*p* < 0.05
*CSN3*	Fwd	Rs110323127	AA GA GG	7 24 46	0.09 0.34 0.57	G A	0.74 0.26	*p* > 0.05
*CSN3*	Fwd	Rs43703015	CC TC TT	6 9 55	0.08 0.24 0.69	T C	0.81 0.19	*p* > 0.05
*CSN3*	Fwd	Rs43703016	AA AC CC	26 34 20	0.34 0.42 0.25	A C	0.54 0.45	*p* > 0.05
*CSN3*	Fwd	Rs43703017	AA GA GG	17 29 34	0.21 0.36 0.42	G A	0.61 0.39	*p* > 0.05

**Table 4 animals-14-01306-t004:** SNP association with milk yield in liters.

Gene	Locus	Genotype	Milk Yield	*p* Value
*CSN1*	rs43703010	AA	8.03 ± 0.18	*p* < 0.05
		AG	9.85 ± 0.13	
		GG	10.09 ± 0.1	
*CSN1S2*	rs109500451	AA	7.55 ± 0.27	*p* < 0.05
		AG	9.21 ± 0.14	
		GG	10.14 ± 0.17	
*CSN3*	rs110079521	AA	10.33 ± 0.05	*p* < 0.05
		AC	8.77 ± 0.11	
		CC	7.6 ± 0.27	
*CSN3*	rs110323127	AA	9.14 ± 0.43	*p* < 0.05
		GA	7.61 ± 0.17	
		GG	9.97 ± 0.08	
*CSN3*	rs43703015	CC	10.72 ± 0.06	*p* < 0.05
		TC	8.43 ± 0.45	
		TT	9.16 ± 0.12	
*CSN3*	rs43703016	AA	10.29 ± 0.06	*p* < 0.05
		AC	9.07 ± 0.14	
		CC	7.6 ± 0.27	
*CSN3*	rs43703017	AA	10.39 ± 0.06	*p* < 0.05
		GA	9.86 ± 0.09	
		GG	7.81 ± 0.15	

**Table 5 animals-14-01306-t005:** Haplotype and their frequency observed in the casein gene.

Haplotype	SNP1	SNP2	SNP3	SNP4	SNP5	SNP6	SNP7	Frequency
GGAGTAA	G	G	A	G	T	A	A	0.2006
AACATCG	A	A	C	A	T	C	G	0.1705
AACGCCG	A	A	C	G	C	C	G	0.0625
AAAGTAA	A	A	A	G	T	A	A	0.0619
AAAGTAG	A	A	A	G	T	A	G	0.0491
AACGTCG	A	A	C	G	T	C	G	0.0482
AGAGTAG	A	G	A	G	T	A	G	0.0472
GGAGCAA	G	G	A	G	C	A	A	0.0366
GACGTCG	G	A	C	G	T	C	G	0.0362
AGCGTCG	A	G	C	G	T	C	G	0.0333
GGAGCCA	G	G	A	G	C	C	A	0.0314
GGCGTCG	G	G	C	G	T	C	G	0.0313
GGAGCAG	G	G	A	G	C	A	G	0.0292
GGAACAA	G	G	A	A	C	A	A	0.025
GGCGTCA	G	G	C	G	T	C	A	0.0218
AGCATCG	A	G	C	A	T	C	G	0.0213
AAAATAG	A	A	A	A	T	A	G	0.0193
GAAGTAG	G	A	A	G	T	A	G	0.0181
AGAGTAA	A	G	A	G	T	A	A	0.0166
AGAATAG	A	G	A	A	T	A	G	0.0139
GGAGTAG	G	G	A	G	T	A	G	0.0109
GGAATAG	G	G	A	A	T	A	G	0.0062
GGAGCCG	G	G	A	G	C	C	G	0.0061
GAAGCAG	G	A	A	G	C	A	G	0.0029
AACACCG	A	A	C	A	C	C	G	0
AGCGCCG	A	G	C	G	C	C	G	0
GGAGTAA	G	G	A	G	T	A	A	0
GACGTCA	G	A	C	G	T	C	A	0

**Table 6 animals-14-01306-t006:** Haplotype association with milk productivity.

Haplotype	SNP1	SNP2	SNP3	SNP4	SNP5	SNP6	SNP7	Frequency	*p*-Value
GGAGTAA	G	G	A	G	T	A	A	0.1964	<0.0001
AACATCG	A	A	C	A	T	C	G	0.1638	<0.0001
AACGCCG	A	A	C	G	C	C	G	0.0625	<0.0001
AACGTCG	A	A	C	G	T	C	G	0.0507	<0.0001
GACGTCG	G	A	C	G	T	C	G	0.0383	<0.0001
GGAGCAA	G	G	A	G	C	A	A	0.0382	0.01
GGAGCCA	G	G	A	G	C	C	A	0.0375	0.1
GGCGTCG	G	G	C	G	T	C	G	0.033	<0.0001
AGCATCG	A	G	C	A	T	C	G	0.0328	<0.0001
GGAACAA	G	G	A	A	C	A	A	0.025	0.0064
GGCGTCA	G	G	C	G	T	C	A	0.0188	0.0062

## Data Availability

The original contributions presented in the study are included in the article, further inquiries can be directed to the corresponding author/s.

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
