# Peer review of "Validation of Bos taurus SNPs for Milk Productivity of Sahiwal Breed (Bos indicus), Pakistan"

_animals, 2024, doi:10.3390/ani14091306_

Round 1

Reviewer 1 Report (Previous Reviewer 2)

Comments and Suggestions for Authors

The revision has undergone significant improvements. I recommended accepting it

Author Response

Many thanks for your valuable comments to improve this manuscript.

Reviewer 2 Report (New Reviewer)

Comments and Suggestions for Authors

Revision of manuscript entitled “Validation of Bos taurus SNPs for Milk Productivity of Sahiwal Breed, Pakistan” 

The introduction is generic; the variants studied are not described; it is reported that they are known in Bos taurus and that they affect milk traits, but it is not clear in which species and breeds the variants were detected, which milk traits are affected by such variants. I think the variants studied should be described in the introduction section. Also the breed is not sufficiently described. The aim of the work must be clearly formulated, preceded by the hypothesis.

The results description is redundant. The research design is not clear: how many animals? which animals did you use for association analysis?

The Discussion is poor; the authors might make more comparisons with Bos taurus, where the variants have already been evaluated, and Bos indicus.

The title reports the experiment was conducted on Bos taurus species, while the simple summary reports Bos indicus. Please correct and check the entire text. Please be consistent throughout the text.

Line 33-34: Please be clear about the traits you take into account, and please be consistent throughout the text. To be clear, only one milk trait has been analyzed in this manuscript, which is milk yield.

Line 35-36. “These above-mentioned SNPs were already reported for significantly up-regulating the caseins protein contents in 36 Bos taurus.” Please rephrase, content is not clear?

Line 66: “effect”, please change to “affect”

Line 78-80: “Conservation of genetic diversity in animal species requires the proper performance of conservation superiorities and sustainable handling plans that should be based on universal information on population structures, including genetic diversity resources among and between breeds”: please rephrase, se sentence is not clear.

Line 87-89: “unfortunately little is known about the indigenous milk producing Zebu cattle breeds of Bos indicus (like Sahiwal breed)”: does the Sahiwal breed belong to the Bos indicus species? Please be consistent throughout the text, including the manuscript title.

“Table 2. Overview of Casein gene SNPs selected for present study”: in Table 2, please eliminate the "Position" column, because it reports the same information for all elements (forward). You can replace it with other information, such as gene region (intron, exon), or SNP position on chromosome 6. The gene names (CSN1S1, CSN3) should also be homogeneous within the Table and throughout the text.

Line 176-177: “Agreement with HWE illustrate the absence  of naturel or artificial selection for these SNPs in our studied population of Bos indicus  (Table 3).” This statement cannot be correct. It is not possible that in a farm aimed at producing milk there is no form of selection, please rephrase.

Line 133-142: you described the protocol used to amplify a fragment via PCR, but how did you reveal the variants? It is not enough to amplify the DNA, please indicate how you revealed the variants, and how many animals were genotyped.

Line 147: “across the genome”: please rephrase, you studied chromosome 6.

Table 3: and 4: please indicate how many animals.

Table 4. “SNP association with milk yield.” please indicate the unit of measurement of milk yield

Line 358: “as milk quantity and quality will be improved”, you only studied milk yield, please rephrase,

Author Response

Many thanks for your valuable comments to improve this manuscript. Point to point author's response to reviewer comments has been attached as a seperate file.

Round 2

Reviewer 2 Report (New Reviewer)

Comments and Suggestions for Authors

The manuscript is much improved, but still cannot be published due to incomplete description of materials and methods section. The PCR procedure used is not completely described and cannot be repeated by an external lab.

Line 141-148: How did you perform genotyping? If all the amplified fragments were 100 bp long, how did you distinguish the variants? In the electrophoresis gel you saw a 100 bp fragment after amplifying a sample of genomic DNA with three primers: how did you assign the genotype? Did the two primers that recognized the SNP have a fluorophore? It would be appropriate to publish some images of the gels used for genotyping.

Please consider adding the species in title, as many researchers don't know the breed and this information migth be helpful. "Bos indicus Sahiwal Breed"

Author Response

Many thanks for your valuable suggestions. Response to Reviewer's report has een attached as seperate file.

Regards,

Round 3

Reviewer 2 Report (New Reviewer)

Comments and Suggestions for Authors

The authors have modified the text, which is now clear.

This manuscript is a resubmission of an earlier submission. The following is a list of the peer review reports and author responses from that submission.

Round 1

Reviewer 1 Report

Comments and Suggestions for Authors

The manuscript received for review deals with an important issue of improving the productivity of native breeds of cattle, such as the Sahiwal breed, classified from Bos indicus. In this work, the authors focused on casein-coding genes. As is well known, caseins are a reservoir of bioactive peptides, regulatory compounds with hormone-like effects that can affect the nutritional value of milk.

In the Introduction part, the authors well justify the need to undertake the research and clearly define the purpose of the research undertaken.

The Materials and Methods part requires some explanation:

From what biological material was the DNA isolated? In line 27 the authors indicate that these are mammary alveolar endothelial cells and in line 93 that they are somatic cells in milk.

In lines 99-107, the authors describe the design of the primers, but do not specify the number and sequence of the primers, which prevents other researchers from reproducing the experiment. I believe that this information should be supplemented, if there is no possibility for it to be in the text of the manuscript, it can be as Supplementary Materials. In the next paragraph, which is a description of the PCR protocol, the authors give annealing temperature of 55°C, but without prior information about the number and sequence of primers, information about the temperature should be verified. Is it one annealing temperature for all primers? Or for individual couples?

In the Results part, Table 1 needs improvement. I believe that since all the studied polymorphisms have been mapped in BTA6, the column indicating the location in the chromosome is redundant; this information can be found in the text. On the other hand, repeating information both in the text (line 128-133) and in the table whether the tested SNPs are missense variant or intronic is unnecessary, i.e. duplication of information.

In Discussion, the authors refer their results mainly to one paper [27] Huang et al., 2012 and this is the paper on Bos taurus cattle. It is a pity that the authors in this part did not use the paper [9] Mohan et al., 2021 (which they already cited in the Introduction anyway) or Singh et al. Genetic polymorphism and association of kappa-casein gene with milk production traits among Frieswal (HF × Sahiwal) cross breed of Indian origin. Iran J Vet Res. 2014 Fall;15(4):406-8. PMID: 27175140; PMCID: PMC4789222.. Both papers mentioned refer to Bos indicus cattle, which probably would add value to the discussion of the results.

There are numerous editorial errors in the text of the manuscript that need to be corrected, e.g.:

Line 18 is: alpha S1 and Alpha S2, should be: alpha S1 and alpha S2

Line 27 is: pure bred Sahiwa, should be: pure breed Sahiwa

And many others like Csn3 (should be CSN3), k-Casein (should be κ-Casein), sometimes uppercase instead of lowercase.

References requires ordering, because, for example, the names of journals are sometimes given as abbreviations, and sometimes whole names, sometimes there is italic and sometimes there is no italic. Comply with the requirements of the journal.

Author Response

Hi,

Many thanks for your valuable comments. Please see the attachment for the response.

With best wishes and kind regards,

Reviewer 2 Report

Comments and Suggestions for Authors

The authors validated in Sahiwal cattle that seven SNPs already reported in casein genes were significantly associated with milk yield. These SNPs can be used as genetic markers for lactation traits in Sahiwal cattle. Overall, this study is meaningful for the molecular breeding of indigenous cattle, but the English language needs to be improved. Before acceptance some points need clarification.

1. There are several places throughout the text where commas are not used in a reasonable manner, resulting in confusing sentence structure, for example, line 44 could be changed to "To enhance milk yield, animal breeders …" and line 52 could be changed to “For the past few years, the SNPs are …”

2. L58: Please change to "Estimating the amount …"

3. L90-92: The number of samples needs to be specified here

4. L92: It is suggested that [21] be changed to the name of the author.

5. L102-103: The primer sequences corresponding to each SNP need to be presented in the form of a table

6. L110-111: The concentration of primers and templates need to be stated here.

7. 125-126: How is the haplotype constructed?

8. L129-130: How did the author come to this conclusion?

9. The names of genes need to be italicized

10. The gene position of the rs43703016 does not seem to be shown in Table 1. In addition, why does the "N" in CSN3 need to be lowercase?

11. The P value for haplotype GGAGCCA in Table 5 was 0.1, which did not reach a statistically significant level. At the same time, it is necessary to explain the specific amount of high milk productivity corresponding to each haplotype.

12. The discussion section should also add the significance of this study

Author Response

(The authors gave the same response as above.)

Round 2

Reviewer 2 Report

Comments and Suggestions for Authors

None